# The Dynamic Coupling Between the Pulse Wake Mixing Strategy and Floating Wind Turbines

Daniel van den Berg[1], Delphine de Tavernier[2], and Jan-Willem van Wingerden[1]

[1]Delft Center for Systems and Control, Delft University of Technology, Mekelweg 2, 2628 CD Delft, The Netherlands
[2]Department of Flow Physics and Technology, Delft University of Technology, Kluyverweg 1, 2629 HS Delft, The Netherlands

**Correspondence:** Daniel van den Berg (d.g.vandenberg@tudelft.nl)

**Abstract.** In recent years control techniques such as dynamic induction control (often referred to as "The Pulse") have shown great potential in increasing wake mixing with the goal of minimizing turbine-to-turbine interaction within a wind farm. Dynamic induction control disturbs the wake by varying the thrust of the turbine over time, which results in a time-varying induction zone. If applied to a floating wind turbine, this time-varying thrust force will, besides changing the wake, change the motion of the platform. In light of the expected movement, this work investigates if applying the Pulse on a floating wind turbine yields similar results to that of the Pulse applied to fixed-bottom turbines. This is done by considering first the magnitude of motions of the floating wind turbine due to the application of a time-varying thrust force and secondly the effect of these motions on the wake mixing. A frequency response experiment shows that the movement of the floating turbine is heavily frequency-dependent, as is the thrust force. Time domain simulations, using a free wake vortex method with uniform inflow, show that the expected gain in average wind speed at a distance of five rotor diameters downstream is more sensitive to the excitation frequency compared to a bottom-fixed turbine with the same Pulse applied. This is due to the fact that at certain frequencies platform motion decreases the thrust force variation and thus reduces the onset of wake mixing.

## 1 Introduction

The drive for the European Union (EU) to become carbon-neutral and energy self-sufficient has increased the demand for renewable energy production. These goals were (re)formulated in the REPowerEU Plan (European Commission, 2022). Offshore wind energy is often regarded as one of the key technologies to provide this green renewable energy (Panwara et al., 2011). As of 2021, offshore wind farms provided 28 GW (13.5%) of the 236 GW of installed wind capacity in Europe. These offshore wind farms are all located in shallow (less than 50 m) waters (Komusanac et al., 2021; Ramirez et al., 2020). However, to achieve the 480 GW of installed wind capacity (on- and offshore combined) target by 2030 (European Commission, 2022), offshore wind needs to find access to the deeper waters of the European Union, and the United Kingdom, where 80% of the total wind energy resources can be found (WindEurope, 2017). Floating wind turbines (FWTs) will play a key role in enabling access to these energy resources. As the technology matures, floating turbines will likely be clustered into large wind farms, similar to the bottom-fixed wind farms in shallower waters.

It is well-known that wind turbines within a wind farm interact with the wakes of surrounding turbines. This results in an extensive reduction of the power production of the individual turbines, which may be in the order of 10 to 25% (Barthelmie et al., 2009, 2010). Since wake interaction is a major source of energy loss, a significant amount of research has been performed on understanding wake effects. In steady design and operational conditions, considerable advances have been made since the first engineering method presented by Jensen (1983). However, particularly in unsteady inflow or operational conditions that are inevitable for floating turbines, wake losses and mitigation techniques remain elusive.

With this work, we will study to what extent the wake losses of floating wind turbines can be minimised using dynamic induction control techniques. In particular, we will look at the Pulse wake mixing technique. More specifically, we will consider if floating wind turbines can leverage their extra degrees of freedom while using this wake mixing strategy.

## 1.1 Background

For bottom-fixed turbines, various strategies have been proposed in the literature in order to reduce wake losses. These strategies include wind farm layout optimization, steady-state control solutions and active wake control strategies (Meyers et al., 2022).

Since the introduction of (engineering) wake models, the placement of wind turbines within a wind farm has received major attention, typically for the goal of wind farm layout optimization. The work by Mosetti et al. (1994) and Petersen et al. (1998) show that optimizing the layout of the wind farm for a given type of turbine, plot size and historic wind information is possible. More recently, Kirchner-Bossi and Porté-Agel (2018) achieved an increase in power production in the order of 2% simulating two real-world wind farms for which the turbines have been redistributed using genetic optimization algorithms. These approaches assume the turbine to be static, where the only degree of freedom is the placement of the turbine with respect to each other. As input to the optimization, historical wind data is used to maximise the potential power gain for the prevailing wind. Because the optimization is static, this potential gain is limited for wind directions that occur less often.

A different approach considers using an optimal steady-state control solution using the available degrees of freedom of a bottom-fixed turbine. The two methods that have gained traction in literature are wake re-direction and induction control. Using the yawing availability of the turbine, the rotor can be statically misaligned with the mean wind direction to divert the wake away from downstream turbines. This increases the power output of downwind turbines at the cost of power loss on the upwind turbine for an overall net gain (Jiménez et al., 2010; Bastankhah and Porté-Agel, 2016; Gebraad et al., 2016). A downside of this method is that, as the number of wind turbines in a farm increases, a diverted wake for one pair of turbines could overlap with a different turbine. Furthermore, the yawed turbine will experience increased loading which reduces the turbine's lifetime (Fleming et al., 2015). Yaw misalignment is an example of *steady-state* optimal control. The use of engineering wake models, such as FLORIS (Gebraad et al., 2014; NREL, 2021), allows for calculating an optimal yaw angle based on measurements within a wind farm. Once the turbine is positioned in the new desired configuration, it is kept steady until the wind conditions change. Recent advances in these engineering wake models have introduced dynamic behaviour (Becker et al., 2022). This allows for optimizing wind farm control under time-varying conditions.

Steady-state induction control (also called de-rating control) is another approach used to increase farm output. It uses the induction factor of the turbine as a control input. Similar to wake redirection, the performance of the first turbine is sacrificed

for the benefit of downwind turbines. Induction control has shown great potential in different simulation and optimization environments (Marden et al., 2013; Ciri et al., 2017; Bossanyi et al., 2022). However, wind tunnel experiments and full-scale experiments have shown that the gain of induction control is negligible (Campagnolo et al., 2016; van der Hoek et al., 2019), contrary to what is found using wake redirection (Campagnolo et al., 2016; Fleming et al., 2017). The work described in Bossanyi et al. (2022) is currently being tested in a real-world wind farm to evaluate the effectiveness of their proposed induction control solution and could potentially yield a different conclusion to Campagnolo et al. (2016) and van der Hoek et al. (2019).

As an alternative to the steady-state optimal control techniques, new forms of active, time-varying, control have gained interest within the scientific community and were first introduced by Goit and Meyers (2015). As a result of this research, two notable active control techniques have emerged: the Pulse (formally Dynamic Induction Control or DIC) (Munters and Meyers, 2016, 2018) and the Helix (formally Dynamic Individual Pitch Control or DIPC) (Frederik et al., 2020a). Both methods rely on disturbing the wake using dynamic blade pitching such that the natural mixing process starts at an earlier distance downstream. For the Pulse wake mixing approach, the blade pitch angle of all rotor blades is varied collectively in a sinusoidal manner. For the Helix strategy, on the other hand, the blade pitch angle of the blades is controlled individually and varies sinusoidally with a phase offset between the blades. Both methods have shown in simulations to increase the power of a two-turbine wind farm by up to 5% for the Pulse and up to 7.5% for the Helix under turbulent inflow conditions (Frederik et al., 2020a).

The proposed techniques to mitigate wake interaction between turbines can also be applied to floating wind turbines. In fact, where bottom-fixed turbines are limited to one degree of freedom (yawing), a floating turbine has the ability to move in all six degrees of freedom. In general, these motions add significant complexity to the design of floating wind turbines, but for wake control purposes, they can potentially be leveraged. Moving turbines could be used for the purpose of repositioning the wind farm, and thus for active layout optimisation. This idea is used by Rodrigues et al. (2015) and Kheirabadi and Nagamune (2020) to actively optimise a wind farm based on wind conditions. In Kheirabadi and Nagamune (2020) several wind farms of differing sizes, placed in a grid layout with a 7-rotor diameter ('$7D$') spacing between turbines, are optimised. In their research, they showed that the overall farm efficiency could be increased by 5-10% by actively optimizing the layout, where the actual percentage of gain depends heavily on wind farm size and wind direction.

Alternatively, for certain types of floater designs, the orientation of the turbine can be changed by changing the ballast. This is done by Nanos et al. (2020), where wake deflection is realised by pitching the platform instead of yawing the turbine. Here, the wake deflects either upwards or downwards, where re-directing the wake downwards showed to increase the overall power production of a two-turbine wind farm (Nanos et al., 2020).

When wake mixing techniques such as the Pulse and Helix are used, the thrust force will vary over time. The research presented in Han and Nagamune (2020) and Kheirabadi and Nagamune (2020) showed that the thrust of the turbine can be used to alter the state of the floating turbine, and the platform to translate and/or rotate. For the Helix wake mixing technique, this is explored in van den Berg et al. (2022). In their work, the yaw moment originating from the Helix is found to primarily excite the yaw degree of freedom of a floating turbine. By enabling the platform motion in yaw during Helix operation, the wind speed downstream increased by up to 10%, compared to a bottom-fixed turbine with the same Helix operation. Furthermore, it

was found that the floating turbine, using the Helix, showed a similar performance as a bottom-fixed turbine, using the Helix with double the pitching amplitude. This gain is thought to originate from the fact that not only the wake is mixed but also dynamically deflected. Further research is required to confirm this statement. For the Pulse wake mixing technique, it is not yet clear if a similar positive coupling between wake mixing technique and floater dynamics exists.

## 1.2   Research objective

The main contributions of this paper are twofold: (i) a frequency analysis of the motions of the floating turbine and its coupling to the Pulse dynamics, and (ii) time domain simulations to investigate the effect of the motions on wake mixing with the system represented by its full non-linear dynamics. These simulations are executed in *QBlade* (Marten, 2022), a simulation suite capable of fully simulating hydro-, aero-elastics and wake dynamics. The wake is modelled using a free-wake vortex model.

The remainder of this paper is organised as follows: Section 2 introduces the simulation tool and settings used in this research. Section 3 gives a short summary of the Pulse wake mixing technology. It also provides insight, using frequency response functions, into the dynamics of a floating turbine when exposed to the Pulse. Section 4 gives time domain results for the floating NREL 5MW turbine with the Pulse at three different frequencies. Finally, Section 5 will present the conclusion of this work.

## 2   Simulation tools and Research Methodology

In this research, *QBlade* is used as the simulation tool (Marten, 2022). *QBlade* uses a free-wake vortex method to simulate the flow field, and thus also the wake, around a turbine. The vortex method is known for its accuracy in the near wake (Leishman et al., 2002, 2004), as well as being computationally more efficient than comparable LES methods (Shaler et al., 2020). However, free-wake vortex methods are prone to numerical instability, especially in the far-wake region. Nevertheless, the vortex method can be used to analyze the wake further downstream, see for example (Marten et al., 2020; Rodriguez et al., 2021). *QBlade* is used as it is able to simulate both the hydrodynamics, as well as the near- to mid-wake. All the simulations will be run with uniform and steady inflow. This provides a best-case scenario for the wake mixing technique. When unsteady inflow is considered natural mixing already occurs in the wake which reduces the effectiveness of the wake mixing techniques. However, wake mixing can still be beneficial even when turbulence is considered (Frederik et al., 2020a).

Throughout this research, without loss of generality, the NREL 5MW turbine (Jonkman et al., 2009) mounted on an OC3 spar-buoy (Jonkman, 2010) is used. Section 4 will touch upon how the results presented in this work would or would not differ for different turbines and different floaters. The OC3 floater with NREL 5MW turbine has been extensively verified against OpenFast calculations and experimental data within the Floatech project (Floatech, 2022).

## 2.1 Numerical set-up

In this section, the settings used in *QBlade* will be motivated. As for any aerodynamic simulation, the chosen settings are a trade-off between computational time and accuracy. In a free-wake vortex method, this trade-off is primarily dictated by the number of vortex elements in the wake, as computational time grows exponentially with the number of wake elements. The settings that influence computational time and accuracy can be divided into two groups. The first directly regulates the number of elements in the wake and is categorised under wake modelling in *QBlade*. In the second group are settings that influence vortex modelling. These settings generally have a larger impact on the accuracy of the wake and less on computational efficiency. Finally, the blade and time step discretization will also influence the number of vortex elements released into the wake and therefore will also have an impact on computational time and accuracy. Figure 1 shows an image of *QBlade* with the fully panelised near wake visible.

**Table 1.** Numerical simulation settings that regulate the wake discretization. These settings are used for all simulations presented in this paper.

| Wake Modeling | Setting |
|---|---|
| Wake Relaxation | 1 (No relaxation) |
| Max. Wake Elements | 200000 [-] |
| Max. Wake Distance | 100 Rotor Diameters |
| Wake Reduction Factor | 0.001 [-] |
| Near Wake Length | 0.5 Revolutions |
| Wake Zone 1/2/3 Length | 6/12/6 Revolutions |

The wake modelling settings are summarised in Table 1. These settings all directly influence the length of the wake and as such also the number of wake elements. The wake is cut-off when either the maximum number of wake elements or the maximum wake distance is reached. Wake relaxation, when enabled, blends the starting vortex and influences the length of the wake. When enabled, the resulting wake is too short for the analysis in this work, hence, for this reason, it is disabled. The wake reduction factor dictates when wake elements are removed based on their current vorticity strength compared to newly released vortices. The final length of the wake is a trade-off mainly between these settings. Throughout the simulations carried out in this work, the wake reduction was the most stringent setting when it comes to the number of wake elements as they were removed before reaching either the maximum number of wake elements or the maximum wake distance.

The full wake is subdivided into four distinct areas. Of this, the near wake is a finely resolved wake close to the turbine which mainly influences the performance of the turbine. Typically, the near wake does not need to be resolved much further than half a rotor diameter for accurate results (Shaler et al., 2020). After the near wake, the wake transitions to wake zone 1. Wake zone 1, and consecutively wake zones 2 and 3, are regions in which the wake is increasingly sparsely resolved. When the wake transitions to a different zone, vortex filaments are merged to reduce the number of elements, which increases computational efficiency. The wake zones are defined in terms of revolutions of the turbine, which, based on the average velocity in the wake,

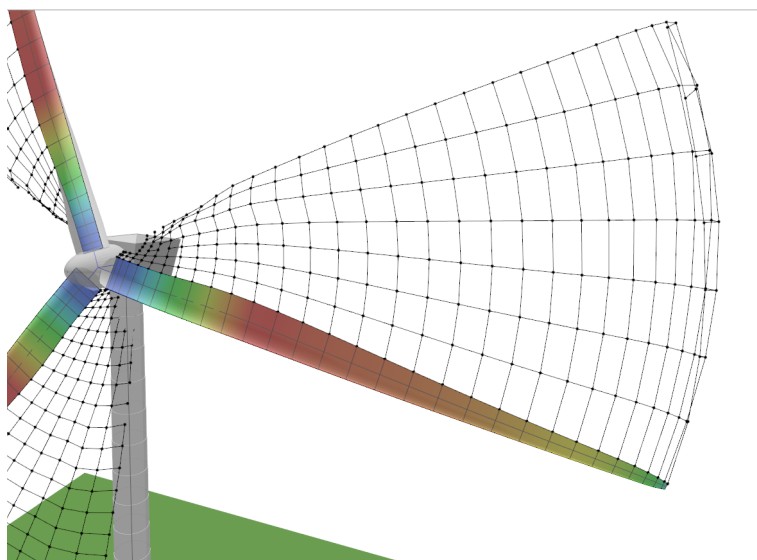

**Figure 1.** A close-up of the discretised blades and near-wake in QBlade. The image is taken at the start of the simulation at which only the near wake is visible. The colour on the blade represents the loading over the blade, with red indicating high load and green/blue lower loading. The vortex elements are represented by the black lines.

**Table 2.** Numerical simulation settings related to the aerodynamic modelling and vortex definition. These settings are used for all simulations presented in this work.

| Vortex Modelling Settings | Setting |
| --- | --- |
| Initial Core Radius | 0.05% Chord |
| Vortex Viscosity | 800 [-] |
| Vortex Strain | Disabled |
| Trailing Vortices | Enabled |
| Shed Vortices | Enabled |

can be translated to a distance. Based on an in-depth grid study, the values in Table 1 were found to give a good trade-off between wake accuracy and computational time. Appendix A provides more detail on the effect of different wake zones on the velocity in the wake.

Settings that influence the vortex modelling are summarised in Table 2. The size of the core radius influences the stability 150 of the wake, the larger the core radius the higher the stability. However, having a too-large core size can also limit the wake mixing dynamics. Vortex viscosity is set to 1100, which was found to work well for modelling large rotors (Ananthan and Leishman, 2004; Berdowski, 2018). Finally, having both trailing and shed vortices increases the accuracy of the wake.

Table 3 summarises key settings for the blade modelling. For each simulation, the time step is set to 0.05 [s]. With an inflow speed of 9 [m/s], the azimuthal step of the turbine is $\Delta\psi = 3.3°$ at rated speed. This azimuthal step was found to be a good

compromise between accuracy and computation time (Shaler et al., 2020). Finally, the blade is discretised in 18 panels, which are sinusoidally distributed over the blade. Appendix A shows, through a convergence study, that this yields accurate results and prevents a large increase in computational time. The simulation is run with the Beddoesh-Leishman unsteady aerodynamic model with the corresponding coefficients based on the research presented in Leishman et al. (1986). These values were validated using data from the MEXICO campaign in (Pereira et al., 2013).

**Table 3.** Numerical simulation settings related to the turbine modelling. These settings are used for all simulations presented in this work.

| Turbine Settings | Setting |
|---|---|
| Dynamic Stall Model | Beddoesh-Leishman |
| Pressure Lag Constant | 1.7 [-] |
| Viscous Lag Constant | 3.0 [-] |
| Time Step | 0.05 [s] |
| Azimuthal Step | $\Delta\psi = 3.3°$ [deg] |
| Inflow Velocity | 9 [m/s] |
| Discretization Panels # | 18 [-] |
| Discretization Method | Sinusoidal |

## 3 The Pulse and Platform Dynamics

This section will summarise the driving principle behind the Pulse wake mixing mechanism. It also provides an analysis of the motions of a floating turbine initiated by the Pulse. This analysis is done based on frequency response functions (Astrom and Murray, 2008). The frequency responses provide insight into how the hydrodynamics of the floating turbine is coupled to the dynamics of the Pulse.

### 3.1 The Pulse Wake Mixing Strategy

The principle of the Pulse is derived from the work of Goit and Meyers (2015) and Munters and Meyers (2018) in which a global optimization for an entire wind farm provided an optimal thrust coefficient $C_t'$ input for the wind turbines with the aim of power maximization. From the optimised wind field, temporal variations in shed vorticity were identified which disrupted the wake. This behaviour in the wake can be mimicked by adding a time-varying offset to the Betz-optimal coefficient, $C_t' = 2$, which can be expressed as Equation 1:

$$C_t'(t) = 2 + A\sin\left(2\pi St\frac{V_\infty}{D}t\right),\tag{1}$$

$$\text{with: } St = \frac{f_e D}{V_\infty},\tag{2}$$

where $A$ is the amplitude [-], $D$ the rotor diameter in [m], $V_\infty$ the inflow velocity in [m/s] and $St$ is the Strouhal number [-], in which $f_e$ is the pitching frequency in Hz. Finally, $t$ is the time in [s]. With this time-varying thrust coefficient, the overall

thrust force on the rotor will vary in a sinusoidal manner. This method only works for free stream turbines and not necessarily waked turbines. If a similar type of actuation is beneficial for waked turbines requires further research. In this paper, we adopt the strategy proposed in (Frederik et al., 2020b). The time-varying thrust force as described in Equation 1 can be realised by pitching the turbine blades with a similar sinusoidal signal.

## 3.2 Floater Dynamics

To capture the behaviour of the OC3 platform (Jonkman, 2010), several frequency response simulations are performed. The Pulse is applied at different frequencies after which the floating turbine reaches steady-state behaviour. Using the steady-state signals, the gain and phase between input and outputs can be mapped.

### 3.2.1 Platform Motions

Figure 2 shows the frequency responses for each of the degrees of freedom of a floating turbine. The turbine response shows

that the type of motion that the floating turbine undergoes is dependent on the excitation frequency, with different movements becoming dominant at different frequencies. For example, while the surge motion is dominant at lower frequencies, the floating turbine undergoes a combination of pitching and surging motions at higher frequencies. All other motions typically appear with at least an order of magnitude lower amplitude than the surge or platform pitching motion. Therefore, they will be neglected in further analysis.

### 190   3.2.2 Effect on Nacelle Displacement

The motion of the nacelle is mainly dependent on the magnitude of and coupling between surge motion and platform pitch, which is a result of floater design (Lemmer et al., 2020). While generally it is expected that a surge and pitch motion results in a nacelle displacement, it is possible that the surge and pitch motion counteract each other, causing the nacelle to remain almost stationary. An example of this is given in Figure 3: it depicts two floating turbines which are both pitching and surging

whilst having different nacelle displacements. As will be shown in the frequency analysis, this is dependent on the phase and gain difference between the two motions. The movement of the nacelle is of interest as it will cause the turbine to experience a time-varying inflow and thus a time-varying thrust loading. It is expected that this may interfere with the working principle of the Pulse. Therefore, the movement of the nacelle is also included in the frequency analysis. The frequency response diagrams for platform pitch, surge and nacelle motion are shown in Figure 4.

At low frequencies, that is until 0.01 [Hz], the motion of the nacelle is nearly one-to-one coupled to the surge motion, as the platform pitch angles are negligible. This can also be seen in the phase data, where the phase of the surge motion and nacelle motion are locked. Between 0.01 and 0.02 [Hz], the nacelle motion is the smallest for the frequency range considered. Within this frequency range the platform surge and pitch motion are almost 180 degrees out of phase. This, combined with

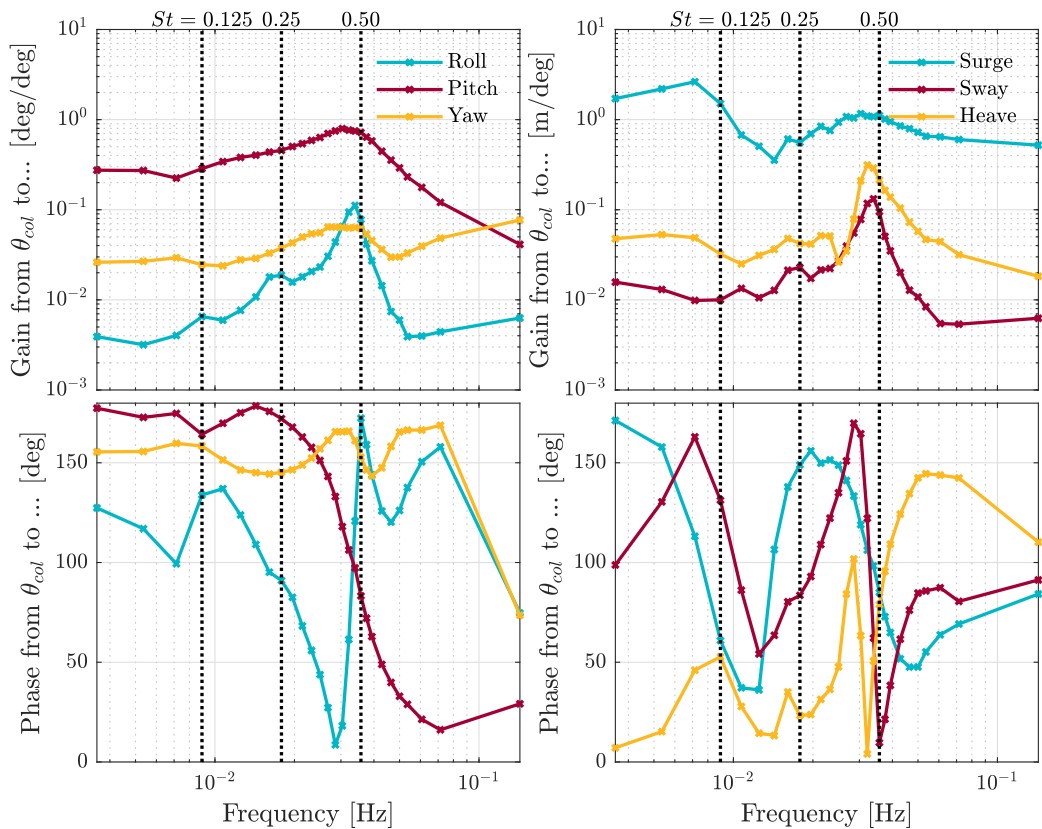

**Figure 2.** Frequency response functions for NREL 5 MW turbine on the OC3 spar-buoy platform. These frequency responses show all translational and rotational motions the turbine will undergo. The vertical dotted lines indicate where blade pitching frequencies at 3 different Strouhal numbers would align with this data. For example, pitching at a frequency of $St = 0.50$ for the NREL 5 MW turbine also means that the system is excited close to the eigenfrequency of the platform pitch motion. In the y-axes label $\theta_{col}$ refers to the collective pitch angle of the blades.

the lower surge motion, results in the floating turbine undergoing pitching and surging motion, without any significant nacelle displacement. This behaviour is depicted in the left situation of Figure 3. At its lowest point, the absolute gain from blade pitch angle to nacelle displacement is only 0.4 [m/deg], meaning for every degree of blade pitch angle the nacelle only moves by 40 centimetres. As frequency increases, the phase difference between surge and platform pitch diminishes and both motions enhance the nacelle motion, graphically depicted in the right situation in Figure 3. This is most prominent at $St = 0.5$, where the nacelle motion achieves its second-to-maximum gain (the first being at a lower frequency).

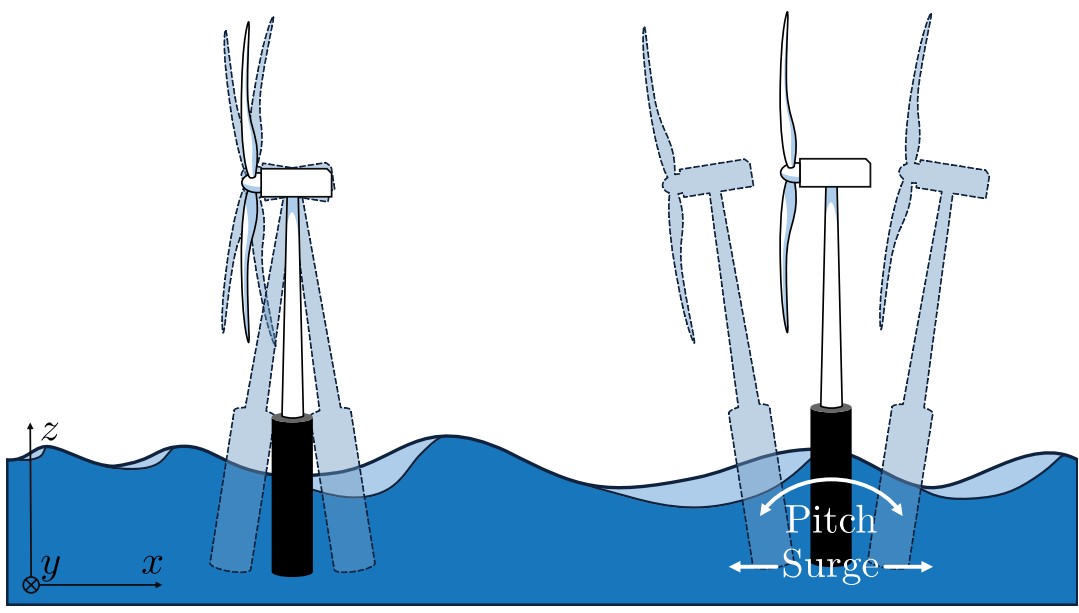

**Figure 3.** Two examples of floating turbines undergoing both surge and pitch motion resulting in different displacements of the nacelle.

### 3.2.3 Effect on Turbine Thrust

Up till this point, only the displacement of the floating turbine at different frequencies has been considered. For a bottom-fixed turbine the frequency, and blade pitching amplitude, at which the Pulse is applied, significantly impact the wake mixing behind the turbine. Typically, the frequency of $St = 0.25$ is taken as ideal when considering two aligned turbines spaced 5 rotor diameters apart (Munters and Meyers, 2018). It could be, however, that this is no longer the case for floating turbines, as the effect of the motion of the floating turbine has to be taken into account. It could well be that actuating at different frequencies on a floating turbine, yields different (local) optima.

One turbine parameter that provides insight into the expected wake mixing is the thrust force. By varying the thrust force, the wake is disturbed through the resulting time-varying wind field. As the turbine moves, it will experience a different relative wind speed, influencing the thrust force of the turbine. It could be that the extra dynamics could lead to locally higher, or lower, peaks in the thrust force variation. Figure 5 shows the frequency response of the thrust force as well as the nacelle motion.

Within Figure 5 two antiresonances can be identified in the floating thrust force. One is located at $0.008$ [Hz] and a more prominent antiresonance is between $0.03$ and $0.04$ [Hz]. At both frequencies there is also a peak in the motion of the nacelle, indicating that there is indeed a coupling between platform and Pulse dynamics. This coupling, however, seems to have a negative impact, as in, as the motion of the nacelle increases the variation in thrust force decreases. The difference in the size of the antiresonance can be explained by considering the frequency at which the motion is at its maximum. The second peak occurs at $\approx 5$ times the frequency of the first peak, which means the nacelle is moving with a higher velocity.

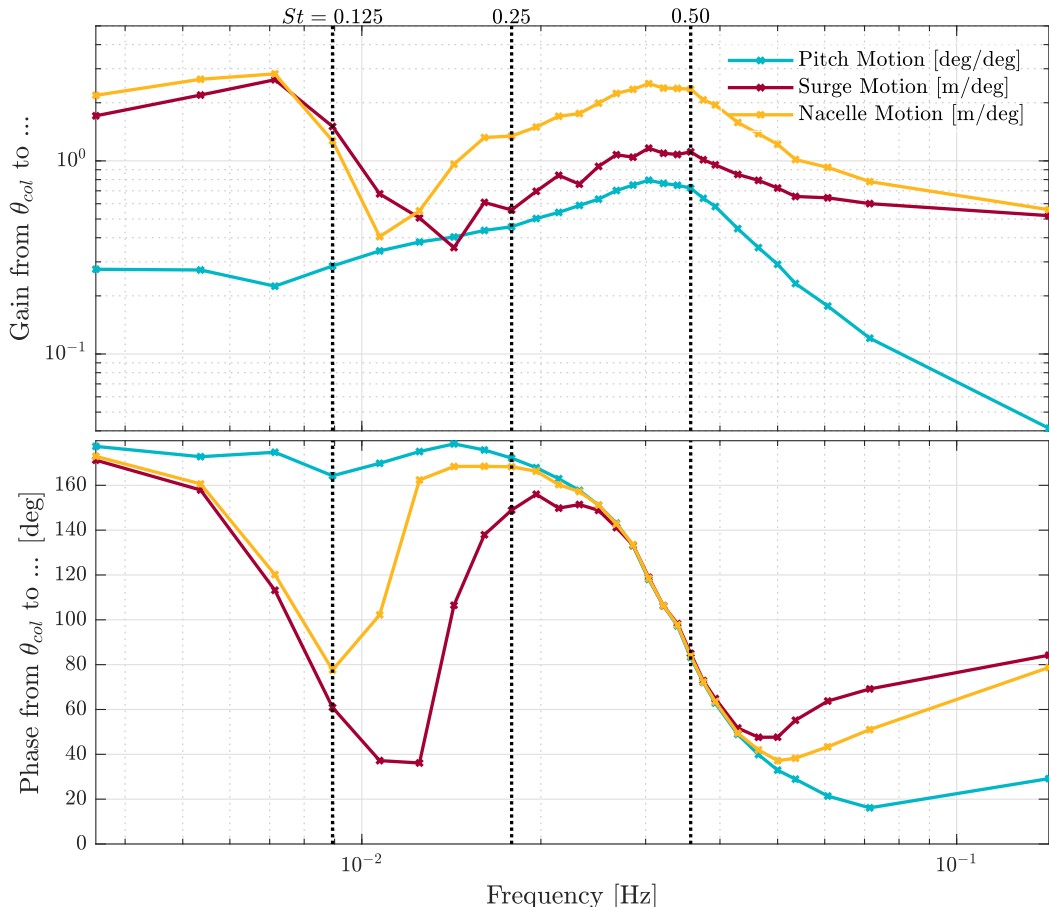

**Figure 4.** Frequency response functions for platform pitch, surge and nacelle motion for NREL 5 MW turbine on the OC3 spar-buoy platform. In the y-axes label $\theta_{col}$ refers to the collective pitch angle of the blades.

At $St = 0.25$ the opposite behaviour to the antiresonance frequencies can be seen. Here, the nacelle motion has an antiresonance and the thrust has a peak. At higher frequencies ($> 0.06$ [Hz]) the gain in thrust reaches its highest gain. However, it is also at this frequency that the Pulse becomes less effective as the blades are pitching too quickly to excite the wake roll-up dynamics. The frequency area of interest is therefore between $St = 0.125$ and $St = 0.5$. Also included in Figure 5 is the gain from blade pitch angle to turbine thrust for a bottom fixed NREL 5MW turbine. At all frequencies, the gain for the bottom-fixed turbine is equal to or greater than that of the floating turbine. At frequencies where the nacelle is not moving, or moving at low velocity, the gain for thrust approaches that of the bottom-fixed turbine. At the resonance peaks of nacelle motion, there is a large difference in thrust force when compared to a bottom-fixed turbine.

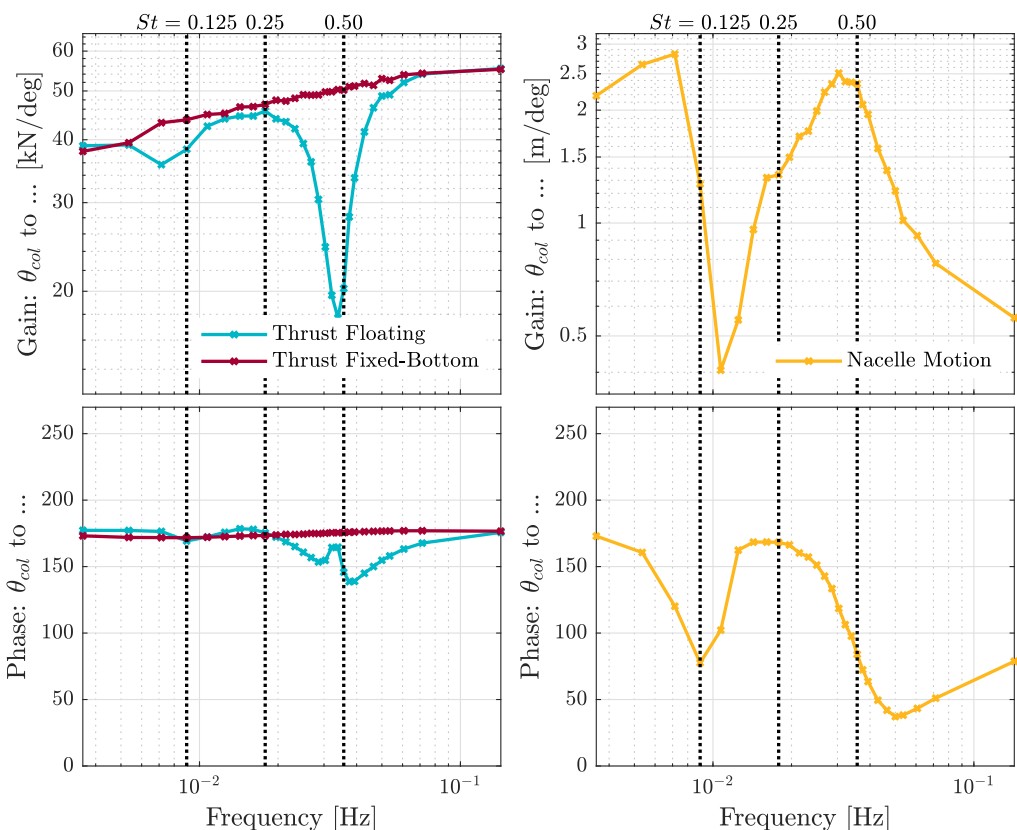

**Figure 5.** Frequency response functions for thrust force variation and nacelle motion, for NREL 5MW turbine on the OC3 spar-buoy platform. The resonance frequencies in nacelle motion coincide with the anti-resonance frequencies in thrust force variation. In the y-axes label $\theta_{col}$ refers to the collective pitch angle of the blades.

The frequency analysis, as presented in this section, is a form of linear system analysis to describe how a system responds to an input signal with one distinct frequency. It fails to capture any non-linear dynamics in the system when the input signal is not a single sinusoid or the amplitude is time-varying. The full, potentially non-linear, behaviour of a floating wind turbine is captured within the time domain simulations. Based on the frequency analysis it is clear that the nacelle motion has an effect on the time-varying thrust force. The full impact on the onset of wake mixing of the wake due to the nacelle motion and the physical displacement of the rotor plane is analyzed in the following section. Based on the thrust analysis it is expected that the movement will affect the degree of wake mixing.

## 4 Time Domain Results

This section investigates if the floating turbine movement for different Pulse frequencies has any noticeable impact on wake mixing. This is done by performing time domain simulations at three different Strouhal numbers: $St = 0.125$, $St = 0.25$ and $St = 0.5$ for both a bottom-fixed and floating turbine. The first and last frequencies correspond to a frequency where the floating turbine shows significant displacement. All floating turbine simulations are compared to a bottom-fixed turbine with the Pulse at the same frequency. First, the time domain results for nacelle displacement and corresponding thrust force will be shown and discussed. Then, for each simulation, the average wind in the wake will be evaluated.

In total 8 different simulations are performed from which it is possible to evaluate downstream wind speeds at different locations. For these simulations, the settings as described in Section 2 are used. In each of the simulations, the turbine is floating unless stated otherwise. The eight simulations are:

1. Constant blade pitch angle (bottom-fixed baseline).

2. Constant blade pitch angle (floating baseline).

3. Pulse with $4°$ pitching amplitude at $St = 0.125$, bottom-fixed.

4. Pulse with $4°$ pitching amplitude at $St = 0.125$, floating.

5. Pulse with $4°$ pitching amplitude at $St = 0.250$, bottom-fixed.

6. Pulse with $4°$ pitching amplitude at $St = 0.250$, floating.

7. Pulse with $4°$ pitching amplitude at $St = 0.500$, bottom-fixed.

8. Pulse with $4°$ pitching amplitude at $St = 0.500$, floating.

### 4.1 Turbine Performance in Time Domain Simulations

Figure 6 shows the nacelle displacement as well as the resulting thrust for the three different frequencies considered. Alongside the data for the nacelle displacement, a simulation with a constant blade pitch angle is depicted for reference. This shows the steady-state position around which the floating turbine is oscillating. For the thrust force data, a reference case with a bottom-fixed turbine excited with the Pulse at $St = 0.25$ is also included. At $St = 0.5$ the nacelle undergoes the largest displacement, with a total movement of 15 metres. As this occurs at the highest actuation frequency, it results in the highest velocity perceived by the nacelle. This combination of large amplitude and high frequency is of greater interest, as it equates to a larger fluctuation in relative wind speed.

The effect of this becomes apparent when looking at the thrust force of the turbine. For $St = 0.5$ the variation in thrust is significantly diminished compared to the bottom-fixed case. This observation is reinforced by looking at the $St = 0.125$ case. For that case, the total nacelle displacement is not significantly less than for $St = 0.5$, but due to the lower frequency, the variation in velocity experienced by the turbine will be substantially slower. This is reflected in the variance of the thrust force,

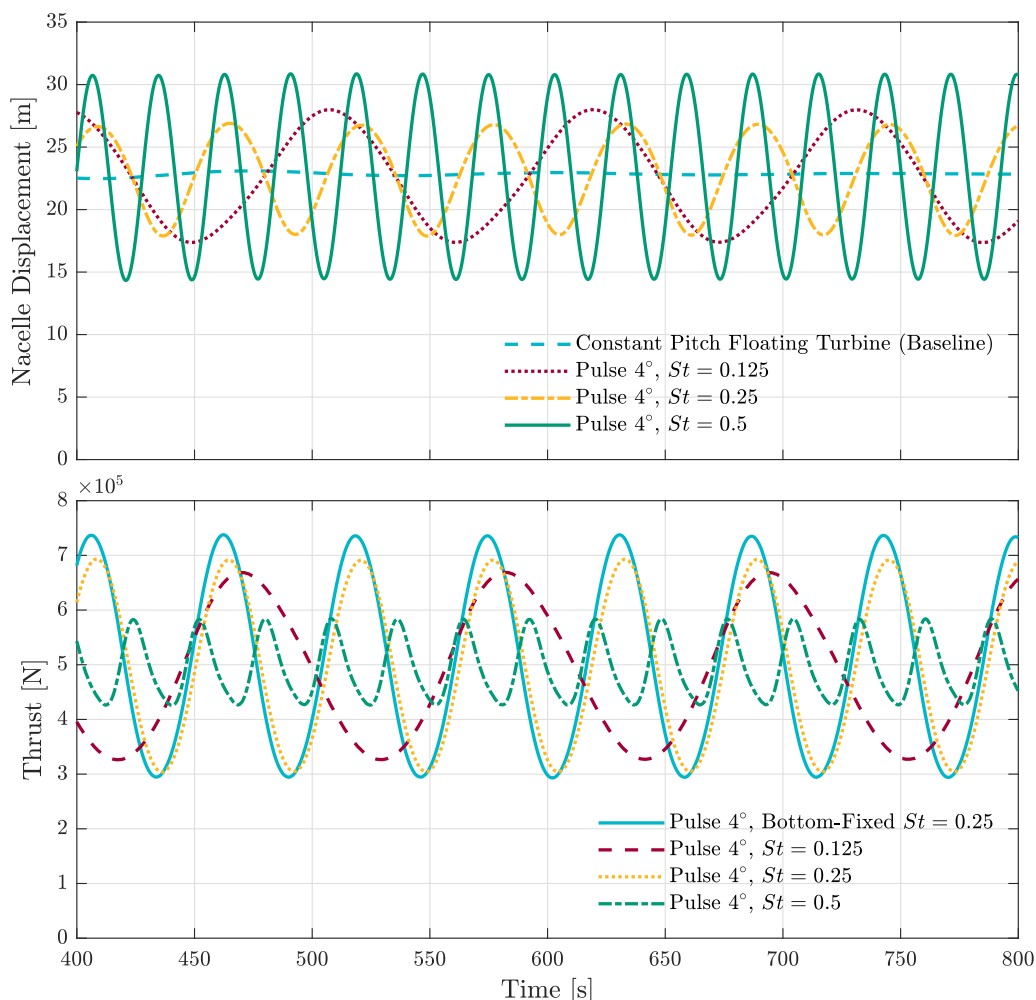

**Figure 6.** Time domain results for platform motion at three different Strouhal numbers. Each frequency has noticeable nacelle displacement which leads to differing behaviour in the thrust force. Note that the cases to which the floating simulations are compared differ between the top and bottom graphs. The top graph shows nacelle displacement with respect to its steady-state un-actuated position, the bottom graph compares thrust force variation to the best-case bottom-fixed simulation.

which is larger compared to the $St = 0.5$ case. As this time-varying thrust force is the driving mechanism behind the Pulse, it is expected that the lower peak-to-peak amplitude results in less wake mixing and thus lower downstream wind speeds.

In conclusion, the lower peak-to-peak amplitude in the thrust force due to the platform motion can be compared to applying the Pulse to a bottom-fixed turbine, but with a smaller amplitude. This is likely reducing the overall effectiveness of the wake-mixing strategy, as investigated in the next section.

## 4.2 Average Wind Speed Downstream

Thus far, the focus has primarily been on turbine performance and its, potential, relation to wake mixing dynamics. The time domain data confirms that the fluctuation of the thrust force varies significantly for different operating frequencies. Up to this point, it is hypothesised that this will affect the degree of wake mixing behind the turbine. This section analyzes the wind speed in the wake, which is directly related to wake recovery due to wake mixing.

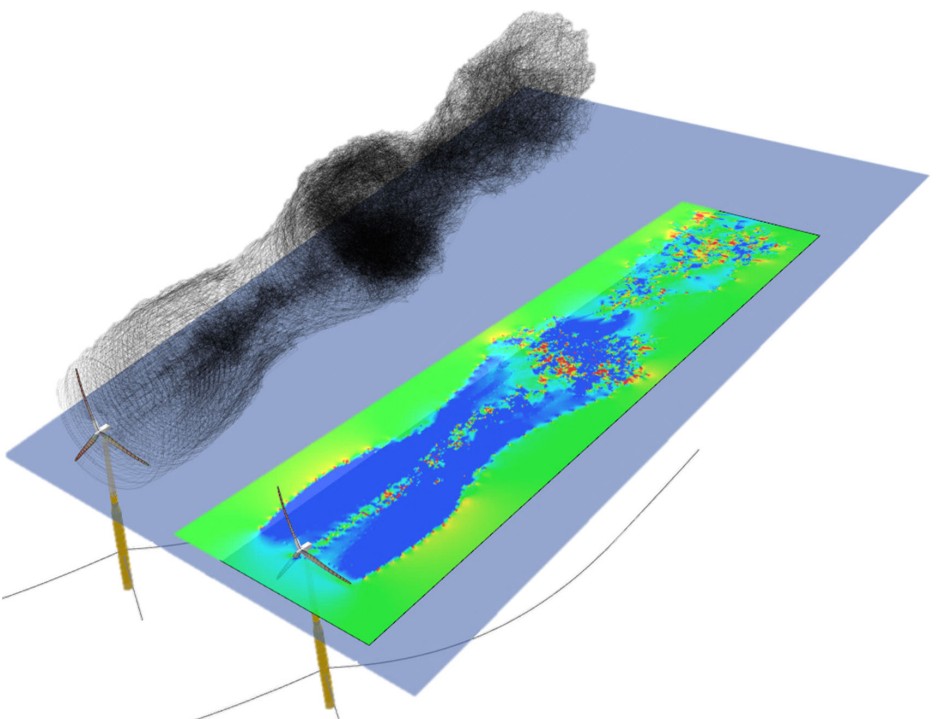

**Figure 7.** A screenshot from QBlade during one of the floating simulations. The left floating turbine shows the wake as represented by the free vortex implementation. Each black line in the wake represents a vortex element. The contraction and expansion of the wake as a result of the time-varying thrust force are clearly visible in the wake. A 2D velocity plane of the same wake is shown on the right-hand side. The bright green colour signifies areas of higher wind speed and blue areas denote areas of low wind speed. Each point in the velocity field is calculated with respect to each of the vortex elements. If a point in the velocity grid is very close to a vortex element it can lead to higher than free stream wind speeds in wake due to the nature of calculating the induced velocity. Such a point is represented by the small red points in the velocity profile.

For each simulation the wind speed is calculated at $0.5D$ distance spacing, starting $0.5D$ in front of the turbine up to $7D$ behind the turbine for a total of 16 velocity measurements. At each $0.5D$ a YZ-plane measuring 360 by 360 metres and centred at turbine level is exported for every 0.5 seconds. This plane is sectioned in squares of 3 by 3 metres yielding a grid of 120 by 120 individual velocity measurements. These dimensions were chosen such that wake expansion is fully captured over the entire domain. For each velocity plane, the average velocity experienced by a second NREL 5 MW turbine downstream,

is computed for each time step for 10 minutes of data (1200 data points per simulation). An example of the wake and its 2-dimensional velocity profile at hub height can be seen in Figure 7.

Figure 8 shows a comparison between cases with the same Pulse frequency and their respective baselines. Between the floating and bottom-fixed baseline there is, when averaged, no difference in downstream wind speed even though there is a slight difference in operating conditions. For the $St = 0.125$ and $St = 0.25$ cases, there is little difference between bottom-fixed and floating cases. At distances of $4D$ and higher, the floating $St = 0.125$ case does show lower wind speeds compared to its bottom-fixed counterpart. However, the most notable difference in downstream wind speed can be seen at $St = 0.5$. Over the entire distance of the considered domain, the wind speed remains lower compared to the bottom-fixed case. As seen in the previous section, it is also the frequency at which the smallest peak-to-peak amplitude in thrust force is seen in the time data.

Figure 9 shows the same data but then all of the bottom-fixed and floating cases are compared with each other. For the bottom-fixed cases, there is a difference in wind speed close to the turbine. As the wake progresses downstream the wind speed converges to similar values between the cases. These findings are different with respect to the work presented in Munters and Meyers (2018) in which there is a difference in energy capture for the different Strouhal numbers. This is likely a result of using a vortex representation to model the wake as it is prone to wake breakdown due to numerical instabilities in the wake. This accelerates the mixing process. For the floating cases, there remains a distinct difference in wind speed, where the $St = 0.25$ outperforms both other pulse cases over the entire domain.

## 4.3 Discussion

Based on the analysis presented in this section it is clear that the additional dynamics of a floating turbine do not necessarily lead to an increase in wake mixing. More importantly, one can conclude that when applying the Pulse whilst disregarding floater dynamics, it could lead to lower performance of the Pulse when applied to an otherwise identical bottom-fixed turbine. In Section 2 it was stated that this work uses the NREL 5MW turbine on the OC3 platform without loss of generality. In this section, a discussion is presented that will elaborate on this statement.

Recent work presented in Broek et al. (2022) confirms the findings in this section. In that work, an adjoint optimization is performed to find the ideal Pulse signal for a 2-turbine floating wind farm, in which only the first turbine is actuated. The optimal excitation signal has a frequency which matches the frequency at which the nacelle is moving the least. The floating turbine in Broek et al. (2022) is modelled as a second-order mass-spring-damper system. Surge motion is represented by a translational mass-spring-damper and platform pitch by a rotational mass-spring-damper system.

In general, the movement of a floating structure/vessel can be modelled as a mass-spring-damper system in which the stiffness and damping properties depend on the hydrodynamic properties and the mooring solution of the floater (Journée et al., 2015). Depending on these values a floating turbine will exhibit eigenfrequencies in pitch and surge, which could result in a similar coupling to nacelle motion as for the OC3 used in this work. As the moment arm from turbine thrust to the centre of gravity of the system is unchanged, the coupling between Pulse dynamics and platform dynamics will remain the same: a change in thrust force will be counteracted by the resulting movement of the platform.

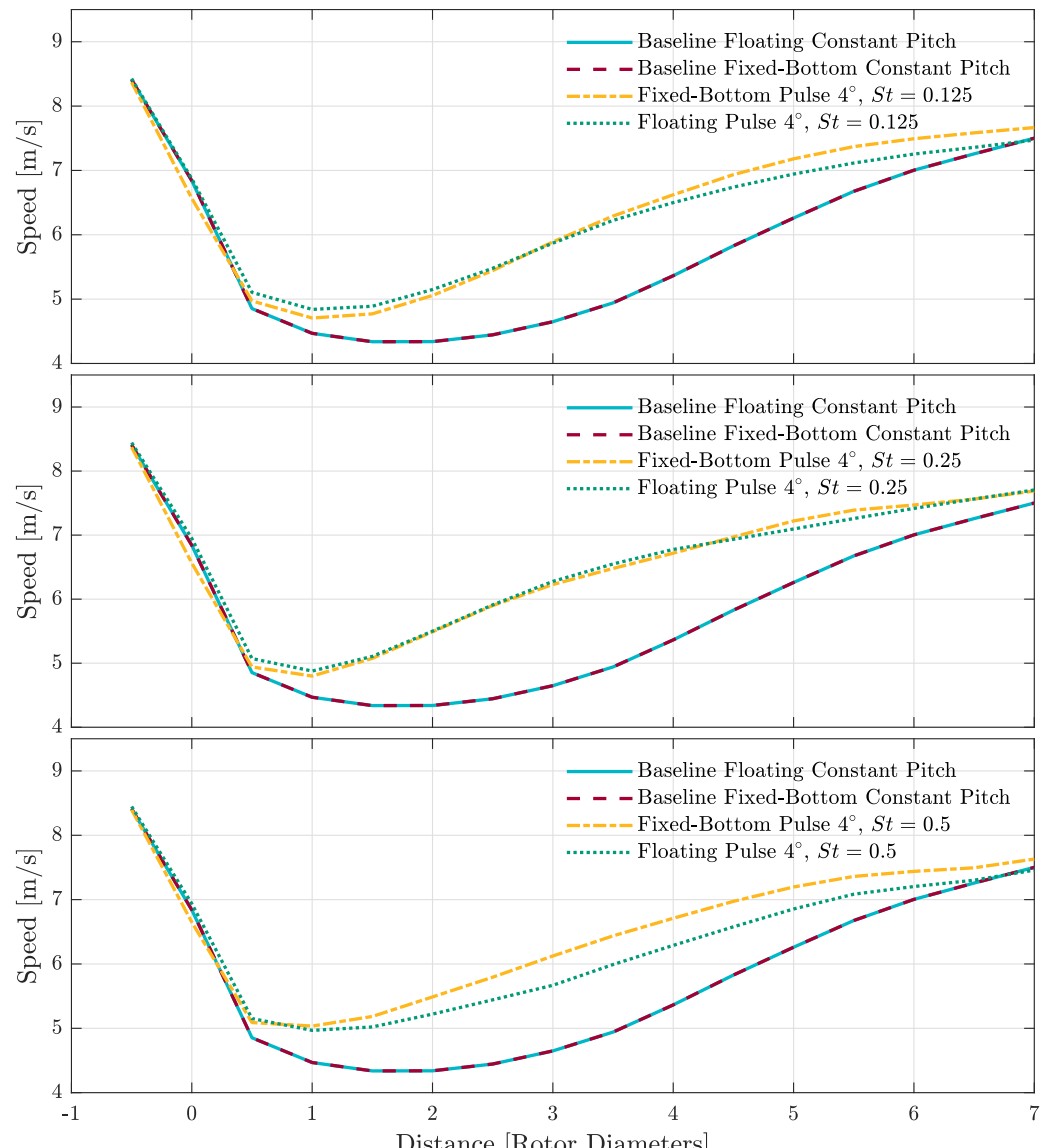

**Figure 8.** Average wind speeds over the analyzed domain for all simulations. Each floating simulation is compared to its bottom-fixed counterpart. Also included in the graphs are both baselines. The top figure shows all the cases for which the excitation frequency is $St = 0.125$, the middle figure has all the cases for $St = 0.25$ and the bottom figure the cases with $St = 0.5$.

The extent to which this will influence the wake mixing technique depends on the floater type and mooring solution. For example, tension leg platforms will show different dynamics to that of a semisubmersible or single spar platform (Journée et al., 2015). The degree to which this coupling exists depends on their respective stiffness. Since floating wind is still a (fast)

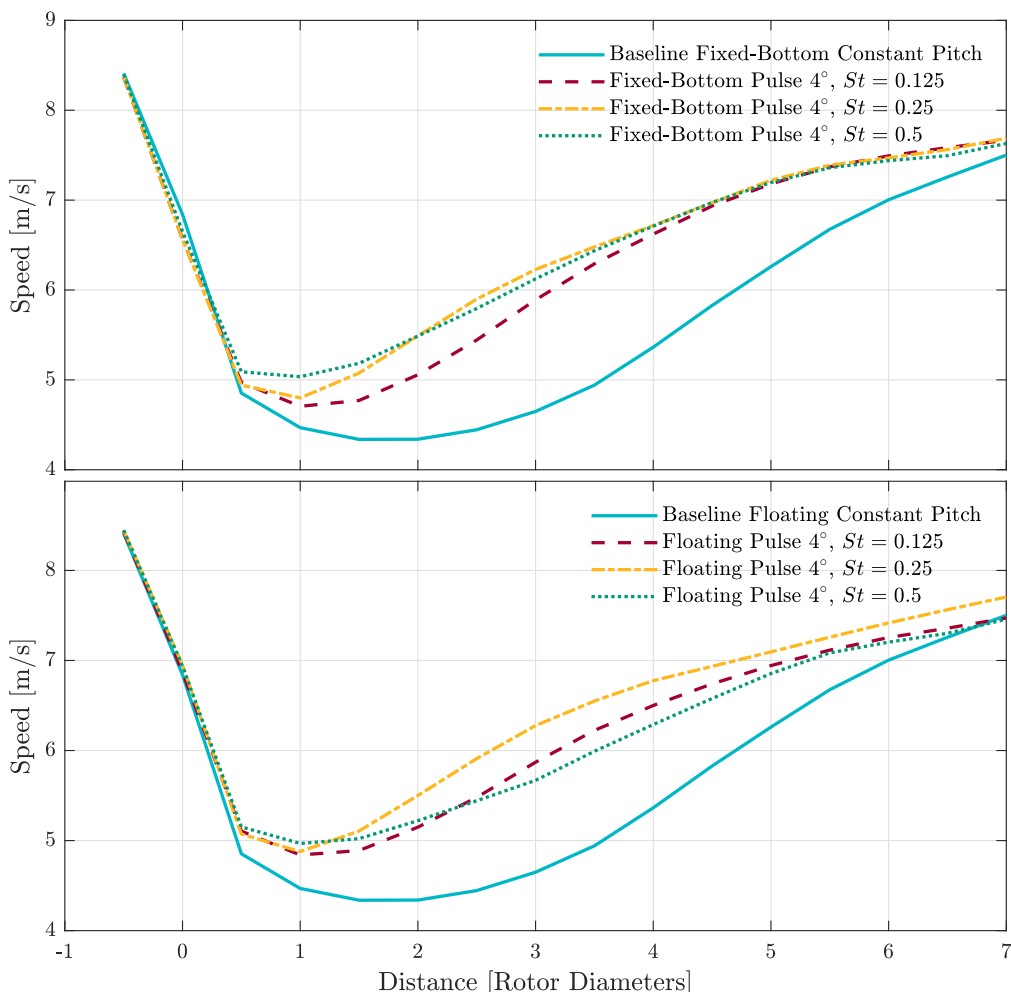

**Figure 9.** Average wind speed comparison between cases with the same mounting method. Where all bottom-fixed cases converge to the same wind speed, a difference between floating cases can be identified. The top graph compares all bottom-fixed cases, the bottom graph compares all floating cases.

emerging technology, a desire to use wake mixing techniques in a floating wind farm could influence floater choice and/or design.

Turbine size could also affect the overall effectiveness of the Pulse wake mixing technique on a floating wind turbine. The Strouhal number is, among other parameters, dependent on the turbine diameter. The larger a turbine is the lower the excitation frequency will be to actuate at a desired Strouhal number. The dynamics of the floating turbine are mainly a result of the hydrodynamic properties of the floater and less so the size of the turbine. The ideal mixing frequency of $St = 0.25$ could

therefore align with the anti-resonance in nacelle motion, leaving the wake mixing technique largely unaffected by the limited platform movement. An opposite scenario is also possible. In such a scenario a different, potentially less effective, excitation frequency should be chosen to get a desirable amount of wake mixing.

It is expected that for different floating turbines the findings in this work would remain largely the same: Given the coupling between floater dynamics and the turbine thrust extra care should be taken when choosing an excitation frequency for the Pulse wake mixing technique.

Finally, it should be mentioned that with dynamic blade pitching techniques such as the Pulse the loading of the turbine will be impacted. Differences in loading might impact the effectiveness of the wake mixing technique. How the potential gains of wake mixing are influenced by the loading of the turbine is a different area of research, see for example (Frederik and van Wingerden, 2022) and (van Vondelen et al., 2023).

## 5   Conclusion

For bottom-fixed turbines, the effectiveness of the Pulse wake mixing strategy depends on both the application frequency as well as the amplitude of the sinusoidal signal. This work shows that the same holds for floating wind turbines. However, the turbine motion induced by the Pulse at different frequencies, predominately in the surge and pitching direction, will further influence its effectiveness and make it more sensitive to the excitation frequency. The dynamic coupling between thrust and the resulting nacelle displacement is such that at certain frequencies the large nacelle displacement results in lower downstream wind speeds, a direct result of a reduction in wake mixing.

This nacelle displacement causes the turbine to experience a varying relative wind speed which negatively impacts the thrust force of the turbine. When the floating turbine is moving due to the Pulse, the movement is such that it lowers the peak-to-peak amplitude of the thrust force. Time domain simulations for three different frequencies show that this lowering of the peak-to-peak amplitude of the thrust force correlates to lower wind speeds downstream in the wake. This implies that the degree to which wake mixing occurs is lowered due to the movement of the floating turbine at certain frequencies.

The work presented in this paper shows that the coupling between wake mixing dynamics and floating turbine dynamics will present a new challenge in finding the right operating frequency, should the Pulse be deployed in a floating wind farm. With this specific floater turbine combination, movement is undesired from a wake mixing perspective. However, new floater designs will also introduce different floating dynamics which might produce different results, or floaters could potentially be designed such that they are guaranteed to enhance wake mixing through their design.

## 6   Acknowledgements

This project is part of the Floatech project. The research presented in this paper has received funding from the European Union's Horizon 2020 research and innovation programme under grant agreement No. 101007142.

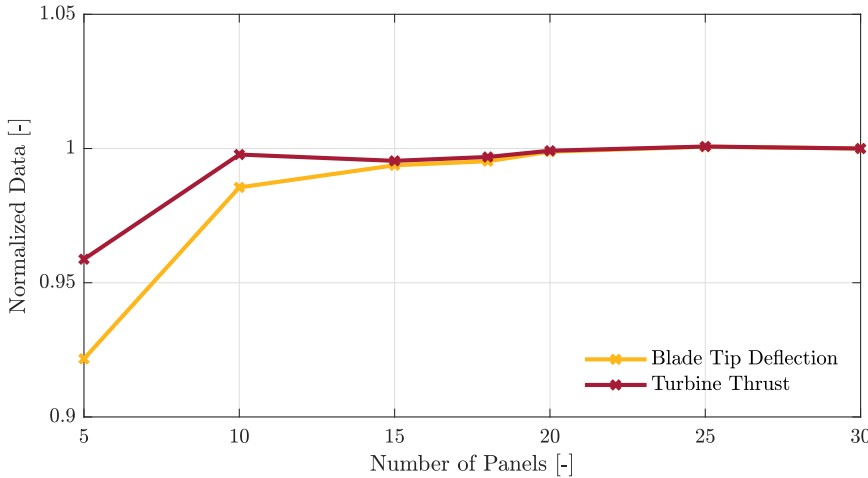

**Figure A1.** Selected results for the convergence results. The plot shows convergence for both an integral parameter (thrust) and an instantaneous parameter (blade tip deflection).

### Appendix A:   Convergence Study

This appendix covers the in-depth convergence study that was done to set the simulation settings used in this research. This convergence study focused on the effect that blade discretization has on turbine performance. Two parameters are analyzed, the turbine thrust and blade tip deflection. The first variable is key to being accurately resolved for the purpose of this research. The second variable is a direct result of the accuracy at which the aerodynamic force is resolved on the blade. Blade discretization also influences the number of vortex elements released into the wake (see Figure 1), which directly impacts computational
time.

    The results of the convergence study are shown in Figure A1. A blade discretization with 30 panels is chosen as the baseline, as it is likely this gives the most accurate representation of the turbine. As will be shown in the results, going higher than 30 yields diminishing marginal gains in terms of accuracy at the cost of computational time. All other data are presented as percentual differences with respect to it. The turbine thrust converges already at 10 panels. The blade tip deflection, however,
converges at 15 panels or higher.

    Furthermore, as the number of panels increases, so does the computational time. Ultimately the choice of the number of blade panels is a trade-off between the desired accuracy and computational time. *QBlade* is provided with a model of the NREL5MW turbine in which the blade is discretised into 18 panels. A data point for 18 panels is also included in the convergence plot.

    The effect of having different wake zones on the wake is also investigated. This is done by looking at the wind speed in
the wake. Whenever the wake transitions from one zone to another, the number of vortex elements is reduced by interpolating among vortex elements and replacing them with a representative, new, vortex elements. Within *QBlade* the length of a wake zone is defined in the number of turbine revolutions. The total number of revolutions for all wake zones summed is kept at

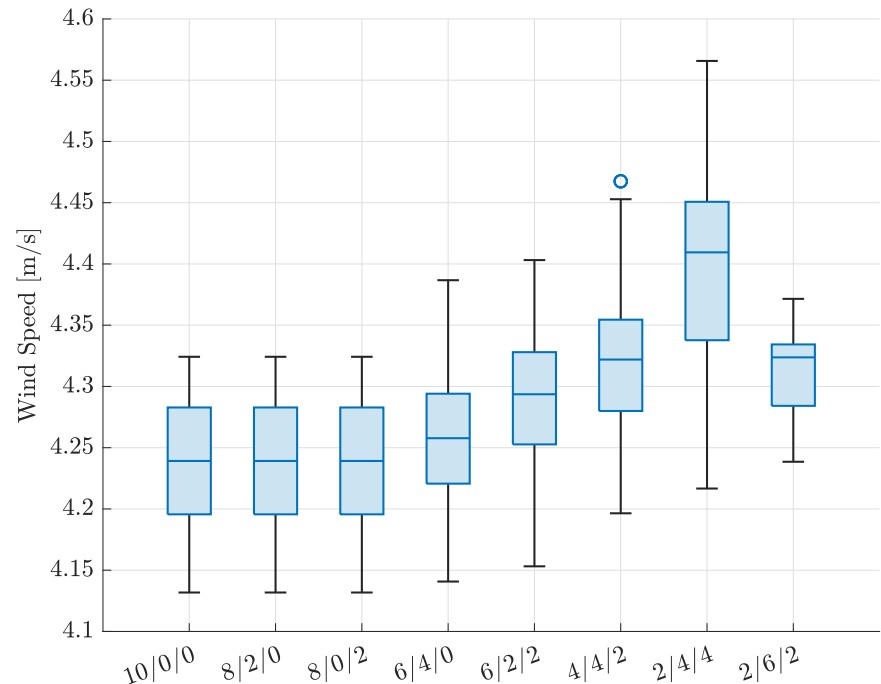

**Figure B1.** Average wind speeds downstream for different wake zones. The numbers on x-axis indicates the unit length of measure for Wake Zone 1/2/3 respectively. The $25^{th}$, $50^{th}$ and $75^{th}$ percentile are shown as a box. The bars indicate the distance to the, non-outlier, minima and maxima in the data set. Any circles indicate data outliers excluded in the boxplot.

10 for this investigation. A further half revolution is reserved for the near wake, which is a fully panelised wake and is visible in Figure 1. The influence of the different distributions of wake zones is analysed by looking at the velocity in the wake at a

distance of 400 metres downstream. The results are presented in Figure B1. From the results in Figure B1, it is clear that there is no difference between cases 1 to 3. This is likely due to the fact that the wake only transitions after the chosen distance of 8 wake zones. This is confirmed by looking at case 4, where the length of zone 2 is expanded. In that case, the velocity field fully falls in zone 2. This results in a slight increase in wind speed and a larger variance which implies that the wake is less stable. Transitioning to wake zone 3 further increases the downstream wind speed and corresponding variance. The biggest takeaway

is that allowing the wake to transition to zone 2 impacts the stability of the wake slightly and results in a small increase in wind speed. However, keeping the velocity field of interest in zone 2 seems to provide a good trade-off between accuracy and computational time. Therefore, the wake zones chosen for this research, as presented in Table 2, are chosen such that the wind field fully falls into zone 2 whilst also allowing zone 1 enough time to fully develop.

*Author contributions.* DVDB set-up and processed all simulations that led to the results presented in this paper under the supervision of
DDT and JWVW. The insights and conclusions presented in this paper are the results of extensive discussions among the three authors.

*Competing interests.* The authors have the following competing interests: At least one of the (co-)authors is a member of the editorial board
of Wind Energy Science.

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
