# Peer review of "The Dynamic Coupling Between the Pulse Wake Mixing Strategy and Floating Wind Turbines"

_Wind Energy Science, 2022_

## Author Response (AR1)

**Reply to the reviewers:**

Dear Reviewers,

We would first like to thank you for the thorough reviews of the submitted paper and feedback. In this document we have gathered the feedback points of both reviewers and explain how it is addressed in the work. The paper has been reviewed by two people. First all comments from reviewer 1 will be addressed after which all comments from reviewer 2 will be addressed. Comments will be headed by *Comment:* the response from the authors is headed by **Response:** and the implementation of the response in the manuscript will be marked by **Action:** after which the changes will be shown.
* * *
**Main comments Reviewer 1:**

*Comment: In the abstract, the authors highlight that the expected gain in average wind speed is more sensitive to excitation frequency for floating than bottom-fixed turbines (this is supported by Figure 9). However, I think an equally of not more important conclusion is that the "gain in average wind speed" (i.e. the enhanced wake recovery) by applying Pulse control is lower for floating than for bottom-fixed turbines due to turbine motions. If the authors agree, this should be added in the abstract.*

**Response:** Thank you for this comment. Based on the results in Figure 9 this is indeed an interesting conclusion. There are, however, two reasons why we chose to omit this from the abstract. From the onset this work is framed to see if the Pulse wake mixing strategy works on a floating wind turbine and if there exists a similar beneficial coupling between platform motion and wake mixing strategy as was seen with the Helix. A direct comparison with a bottom-fixed turbine wasn't the primary focus and the main author feels that adding this to the abstract will distract from the previously mentioned goal of the work.

It is also difficult to directly compare a floating turbine and bottom-fixed turbine. This is mainly due to the fact that in steady state a floating turbine will have a different rotor tilt angle compared to a bottom-fixed turbine, because of the platform pitch angle. This causes a upwards/downwards deflection of the wake, which increases downstream wind speed in the direct stream tube behind the turbine. This can potentially skew the results in favour of a floating or bottom-fixed turbine.

*Comment: In line 111, the authors claim that, "without loss of generality, the NREL 5MW mounted on an OC3 spar-buoy is used". However, in line 179, they also state that "the motion of the nacelle is mainly dependent on the floater design". These statements appear to contradict. Can the authors comment on to which extent their results can be extrapolated to (1) larger turbines representative of current and near-future offshore wind turbines (2) Different floater concepts. I expect it is difficult to generalize this, and would invite the authors to comment on this in the manuscript.*

**Response:** This is a very interesting remark, thank you for this point. Trying to formulate a response to this proved to be valuable for the overall quality of the paper and I personally enjoyed working on an answer for the work. First with respect to the sentence in line 179, it refers to how surge and platform pitch motion will couple (not written in the original sentence!), and that different floater designs will lead to different coupling frequencies. This might influence the motion of the nacelle when the platform is undergoing surging and pitching. We will clarify this in that sentence.

Regarding point (1), different turbine sizes without altering the floater concept will be subject to the same coupling as described in the work. However, as rotor diameters increase in size, excitation frequencies for the Pulse decrease. It could well be that the ideal mixing frequency falls together with one of the eigenfrequencies of the floating turbine. If such a situation occurs it would doubly negatively impact the effectiveness of the Pulse because to reduce the wake interaction between turbines a different excitation frequency has to be chosen. However, this excitation frequency will be less effective than the ideal mixing frequency.

Regarding point (2): A similar story holds for different floater designs, but it is more difficult to comment on since there are different floater concepts. That being said it is still conceivable that the conclusions found in this work will hold for a large collection (possibly all) of floater solutions. The dynamics for any floating vessel can be modelled as a mass-spring-damper system. The mass, spring stiffness and damping coefficient become a function of the hydrodynamics properties of the floating system.

The moment-arm between to the turbine thrust to the CoG of the floating turbine will not change depending on floater design. This means that the moment that is causing the pitching will always be the same, and the direction of pitching will also always be the same. The same will hold for the surge motion, which is a directly related to the thrust force varying over time. If the thrust force increases the platform will be pushed backwards and vice versa.

Even though the basic mechanics will be the same, the magnitudes and locations of the resonances and anti-resonance in the tower top motion **will** be dependent on the platform design. The best way to mitigate this coupling would be to have an ultra-stiff floating turbine, but that would imply bottom-fixed solutions. I would even argue that a similar coupling described in this paper exists for bottom-fixed turbines. However, since the pitch stiffness is significantly higher, and a bottom-fixed turbine cannot surge, it has no noticeable impact on the Pulse performance.

In light of this comment a new subsection has been added to Section 4. This section is titled "Discussion on Results".

**Action:** After *"without loss of generality, the NREL 5MW mounted on an OC3 spar-buoy is used"* the sentence "Section 4 will touch upon how the results presented in this work would or would not differ for different turbines and different floaters." has been added.

Line 179 has been changed from: *"The motion of the nacelle is mainly dependent on the floater design (Lemmer et al., 2020)."* To: *"The motion of the nacelle is mainly dependent on the magnitude of and coupling between surge motion and platform pitch, which is a result of floater design (Lemmer et al., 2020)."*

The following paragraphs have also been added to the end of the results Section:

"Based on the analysis presented in this section it is clear that the additional dynamics of a floating turbine do not necessarily lead to an increase in wake mixing. More importantly, one can conclude that when applying the Pulse whilst disregarding floater dynamics, it could lead to lower performance of the Pulse when applied to an otherwise identical bottom-fixed turbine. In Section 2 it was stated that this work uses the NREL 5MW turbine on the OC3 platform without loss of generality. In this section, a discussion is presented that will elaborate on this statement.

Recent work presented in Broek et al. (2022) confirms the findings in this section. In that work, an adjoint optimization is performed to find the ideal Pulse signal for a 2-turbine floating wind farm, in which only the first turbine is actuated. The optimal excitation signal has a frequency

which matches the frequency at which the nacelle is moving the least. The floating turbine in Broek et al. (2022) is modelled as a second-order mass-spring-damper system. Surge motion is represented by a translational mass-spring-damper and platform pitch by a rotational mass-spring-damper system.

In general, the movement of a floating structure/vessel can be modelled as a mass-spring-damper system in which the stiffness and damping properties depend on the hydrodynamic properties and the mooring solution of the floater (Journée et al., 2015). Depending on these values a floating turbine will exhibit eigenfrequencies in pitch and surge, which could result in a similar coupling to nacelle motion as for the OC3 used in this work. As the moment arm from turbine thrust to the centre of gravity of the system is unchanged, the coupling between Pulse dynamics and platform dynamics will remain the same: a change in thrust force will be counteracted by the resulting movement of the platform.

The extent to which this will influence the wake mixing technique depends on the floater type and mooring solution. For example, tension leg platforms will show different dynamics to that of a semisubmersible or single spar platform (Journée et al., 2015).The degree to which this coupling exists depends on their respective stiffness. A similar coupling will likely exist for bottom-fixed turbines. However, since its `pitching' and especially `surging' stiffness is significantly larger than that of a floating turbine, it will not have a noticeable impact on the effectiveness of the Pulse. Since floating wind is still a (fast) emerging technology, a desire to use wake mixing techniques in a floating wind farm could influence floater choice and/or design.

Turbine size could also affect the overall effectiveness of the Pulse wake mixing technique on a floating wind turbine. The Strouhal number is, among other parameters, dependent on the turbine diameter. The larger a turbine is the lower the excitation frequency will be to actuate at a desired Strouhal number. The dynamics of the floating turbine are mainly a result of the hydrodynamic properties of the floater and less so the size of the turbine. The ideal mixing frequency of St=0.25 could therefore align with the anti-resonance in nacelle motion, leaving the wake mixing technique largely unaffected by the limited platform movement. An opposite scenario is also possible. In such a scenario a different, potentially less effective, excitation frequency should be chosen to get a desirable amount of wake mixing.

It is expected that for different floating turbines the findings in this work therefore would remain largely the same: Given the coupling between floater dynamics and the turbine thrust extra care should be taken when choosing an excitation frequency for the Pulse wake mixing technique."

*Comment: The authors refer to the work of Munters and Meyers (2018) (I will refer to this as MM18), where the Pulse concept was first identified. However, some important differences that complicate a direct comparison (e.g. on optimal excitation frequencies and impact on wake recovery) should be kept in mind which are not addressed in the paper. It would be useful for authors to comment on these in the paper.*

**Response:** Thank you for this comment. We believe that this comment will be answered by addressing the following two comments:

*Comment: The work of MM18 considers a turbulent boundary layer inflow, whereas the current work based on Qblade-Ocean appears to consider a uniform and steady inflow (although it is not explicitly mentioned). This significantly impacts natural wake recovery. Please comment on this and potentially mention it as an important area for future work.*

**Response:** Thank you for attending us that this distinction is not made in the current work. Uniform inflow produces a much more stable wake and thus wake mixing techniques will be significantly more impactful. When unsteady inflow is considered natural mixing will already be present which will reduce the effectiveness of the wake mixing technique. This has been researched for the Pulse and Helix in Frederik et al., 2020a for bottom-fixed turbines. It is expected that the results in that paper considering turbulent inflow carry over to the work presented in this paper and that it doesn't affect the dynamics described. A passage mentioning the use of uniform inflow has been added.

**Action:** The following passage has been added to the introduction of Section 2: "All the simulations will be run with uniform and steady inflow. This provides a best-case scenario for the wake mixing technique. When unsteady inflow is considered natural mixing already occurs in the wake which reduces the effectiveness of the wake mixing techniques. However, wake mixing can still be beneficial even when turbulence is considered (Frederik et al., 2020a)."

**Comment:** *L285: The authors mention that their finding of "at a 5D distance, the wind speed has converged to nearly identical values." aligns with those in Munters and Meyers (2018). However, Fig. 17a in MM18 clearly shows that overall power gains remain quite different for the different St values of 0.125, 0.25, and 0.5 presented here. Therefore, I do not follow this statement, please explain.*

**Response:** Rather than aligning with MM18 it is actually, as commented, a different finding to MM18. Thank you for pointing out this inconsistency. The difference between MM18 and this work is likely the result of using a free-vortex representation for the wake as it is prone to computational mixing (i.e. mixing due to numerical instabilities in the solver), especially when the wake is already unsettled due to the mixing. If the results were to align with MM18 one would expect differences in wind speed at 5D. This will be addressed in the manuscript, including a possible explanation as to why this difference exists.

**Action:** The passage mentioned in the comment has been changed to: "These findings are different with respect to the work presented in MM18 in which there is a difference in energy capture for the different Strouhal numbers. This is likely a result of using a vortex representation to model the wake as it is prone to wake breakdown due to numerical instabilities in the wake. This accelerates the mixing process. For the floating cases, there remains a distinct difference in wind speed, where the St=0.25 outperforms both other pulse cases over the entire domain."

**Minor / typographical comments Reviewer 1:**

**Comment:** *In table 1, the maximum wake length is shown as "100 Rotor Diameters". In the accompanying text, the authors indicate that the wake is cut off either when this length is exceed, or the maximum number of wake elements is reached. Does this setup length of 100 diameters effectively mean that the latter condition is always the most stringent? If not, 100 Rotor diameters appear excessive for the purposes of this paper. Can the authors shortly comment on this?*

**Response:** This is indeed the case with these values. Depending on number of wake elements originally created, which are also influences by settings, it can become restrictive. When setting up the simulation 100 Rotor Diameters is a default setting and kept at that value. There is no particular reason for it. A short text on the trade-off between these settings will be added.

**Action:** The following passage has been added after table 1: "The final length of the wake is a trade-off mainly between these settings. Throughout the simulations carried out in this work,

the wake reduction was the most stringent setting when it comes to the number of wake elements as they were removed before reaching either the maximum number of wake elements or the maximum wake distance."

*Comment: Page 7, L 157: Can the authors add a short comment that, as described in MM18, the Pulse only works for freestream turbines, and that, for waked turbines further research is needed?*

**Response:** A sentence will be added to Section 3.1 to address this comment.

**Action:** The following sentences have been added: "This method only works for free stream turbines and not necessarily waked turbines. If a similar type of actuation is beneficial for waked turbines requires further research."

*Comment: Page 11, L216. "the Pulse becomes less effective as the blades are pitching at a too high frequency" add why the high frequency causes an ineffective Pulse, e.g. since the wake roll-up dynamics are not receptive to these high frequencies.*

**Response:** Thank you for this suggestion, this statement will be clarified.

**Action:** The sentence has been changed too: "… blades are pitching too quickly to excite the wake roll-up dynamics".

*Comment: Figure 4, bottom panel: I assume the ylabel should be "Phase" and not "Gain".*

**Response:** Thank you for spotting this! It should have been phase rather than gain. This will be changed.

*Comment: Page 14, L268: "This section analyzes the wind speed in the wake, another measure for wake mixing", I would suggest a different phrasing, as the wind speed in the wake is the direct product of enhanced wake recovery / wake mixing, not just another measure. Related, I suggest the authors amend the statement at the end of section 4.1 as "This is likely reducing the overall effectiveness of the wake mixing strategy, as investigated in the next section".*

**Response:** This is a nice suggestion, and it strengthens the work presented. The direct relation between wake mixing and the wind speed in the wake will be clarified in those sentences.

**Action:** The two sentences have been changed to: *"This section analyzes the wind speed in the wake, which is directly related to wake recovery due to wake mixing"* and *"This is likely reducing the overall effectiveness of the wake-mixing strategy, as investigated in the next section."* Finally, in the conclusion the last sentence of the first paragraph has been changed to: "The dynamic coupling between thrust and the resulting nacelle displacement is such that at certain frequencies the large nacelle displacement results in lower downstream wind speeds, *a direct result of a reduction in wake mixing.*" With the italic part being added.

*Comment: Figure 7 makes mention of a velocity profile for visualization, yet the figure does not show profiles, but a wake visualization and a planar color plot. Can the authors describe what is exactly shown in the figure? (e.g. the locus of wake elements on the left and contours of axial velocity on the right?)*

**Response:** Thank you for this comment. It lead us to look at all the captions in the work with this comment in mind. We believe that the figures are much clearer now.

**Action:** The caption has been expanded to: "A screenshot from QBlade during one of the floating simulations. The left floating turbine shows the wake as represented by the free vortex implementation. Each black line in the wake represents a vortex element. The contraction and expansion of the wake as a result of the time-varying thrust force are clearly visible in the wake. A 2D velocity plane of the same wake is shown on the right-hand side. The bright green colour signifies areas of higher wind speed and blue areas denote areas of low wind speed. Each point in the velocity field is calculated with respect to each of the vortex elements. If a point in the velocity grid is very close to a vortex element it can lead to higher than free stream wind speeds in wake due to the nature of calculating the induced velocity."

*Comment: Some of the captions can be a little more descriptive to aid the reader to distinguish between plots. For example, in Figure 8, can authors please add in the caption that the top panel is at St = 0.125, etc. It is included in the labels, but mentioning this in the caption makes it easier to distinguish.*

**Response:** Captions will be looked at across the paper and points like these will be considered, see also the previous point.

**Action:** The mentioned caption has been expanded to: "Average wind speeds over the analyzed domain for all simulations. Each floating simulation is compared to its bottom-fixed counterpart. Also included in the graphs are both baselines. The top figure shows all the cases for which the excitation frequency is St=0.125, the middle figure has all the cases for $St=0.25$ and the bottom figure the cases with St=0.5."

*Comment: The authors comment on computational cost / time a few times throughout their paper, especially when referring to choices in the simulation setup. Can they illustrate this by mentioning an order of magnitude for the QBlade-Ocean frequency and time-domain simulation times?*

**Response:** This would be difficult to compare with alternative methods, since computation time is heavily dependent on simulation settings. For the simulations carried out in this work, a single simulation to generate the downstream wind speeds typically took around 10 hours to complete. Post-processing the data typically took 1-2 hours. Comparable LES simulations would probably take in order of a week to run. The frequency response simulations used unsteady BEM and were all run over the span of a night.
* * *
**General Comments Reviewer 2:**

*Comment: A first general comment I have is that I might propose to change the framing of the paper, although maybe this is too subtle a difference to matter too much. My thinking was in the introduction section I was understanding an objective of the paper was to look for benefits of implementing pulse on a floating platform, and in that light the conclusion might appear like a negative result. But in my mind an alternative framing could be that, this and a few other papers are interested if it's even possible to do this type of control on a floating platform. And here the conclusion is more positive, it is possible but on floating it's important to account for the coupling dynamics to motion.*

**Response:** This is an interesting point and in my eyes shows the great value of in depth reviews for journal papers. The framing of the paper has been a point of discussion between the authors throughout the entire writing process. The framing proposed in the comment is

also the desired framing and getting this 'first time reader' insight shows that it might be good to apply subtle changes to get to the desired framing. In hope of achieving this, I have changed two sentences in the abstract under the assumption that an interested reader will read this first. I hope that this will have the desired effect.

**Action:** The final sentence of the abstract has been changed to: "This is due to the fact that *at certain frequencies* platform motion decreases the thrust force variation and thus reduces the onset of wake mixing." The following sentence has been changed from: "This work investigates if the coupling between the Pulse and floater dynamics has an impact on the wake mixing performance of the Pulse." To: "In light of the expected movement, this work investigates if applying the Pulse on a floating wind turbine yields similar results to that of the Pulse applied to fixed-bottom turbines."

*Comment: A second general comment is that I anticipate there will be questions about the loading impacts. Here I think it could be useful also to frame this paper as confirming the conditions under which it is possible to implement this type of control on a floating platform and realize an uplift. Perhaps this won't be true for every type of wind farm control, and then loading analysis can follow on those methods that pass this first screening.*

**Response:** Thank you for this comment, typically these kind of active wake mixing papers are accompanied with questions about loadings ☺. As of now this is still very much an open question. Work with regards to the loading of the Helix can be found in: "Frederik, Joeri Alexis and van Wingerden, Jan-Willem, On the Load Impact of Dynamic Wind Farm Wake Mixing Strategies" at https://dx.doi.org/10.2139/ssrn.3910237.

The Pulse wake mixing technique generates a cyclic loading on this turbine which might increase loading. However, the frequency of loading is also low, so it might not be consequential. Add to this that w.r.t. bottom-fixed turbine loading of the floater also has to be taken into account, which increases the complexity of this question. Since it is too unclear to make any comments on the impact of loading it is left unmentioned in this work.

**Specific Comments Reviewer 2:**

*Comment: Page 2: "Yaw misalignment is an example of steady-state optimal control. The use of a model, such as FLORIDyn (Becker et al., 2022; Gebraad and van Wingerden, 2014)". Wouldn't it make sense to refer to FLORIS rather than FLORIDyn if discussing steady-state control design?*

**Response:** Yes this would for sure make more sense, thank you for pointing this out. I will include a direct reference to FLORIS (which is also the Gebraad and van Wingerden, 2014 reference) in the text.

**Action:** The following change was implemented:" The use of engineering wake models, such as FLORIS (Gebraad and van Wingerden, 2014; NREL, 2021), allows for calculating an optimal yaw angle based on measurements within a wind farm. Once the turbine is positioned in the new desired configuration, it is kept steady until the wind conditions change. Recent advances in these engineering wake models have introduced dynamic behaviour (Becker et al., 2022). This allows for optimizing wind farm control under time-varying conditions."

*Comment: "However, wind tunnel experiments and full-scale experiments have shown that the gain of induction control is negligible (Campagnolo et al., 2016; van der Hoek et al., 2019), contrary to what is found using 60 wake re-directing (Campagnolo et al., 2016). Similar results were found in a full-scale experiment (Fleming et al.,2017)."*

*2 quick comments, I believe there is some newer work on this topic that might indicate some potential afterall (https://iopscience.iop.org/article/10.1088/1742-6596/2265/4/042032/meta ) . Also, I don't think Fleming 2017 includes a test of induction control.*

**Response:** Thank you for pointing our this source. I have read through the work and I feel like, as with the other cited works, it shows potential for axial induction control in simulations (Similar to the already mentioned sources). The work mentioned also describes preparatory work for a field test with induction control and it remains to be seen what is found in that field test.

Regarding the second comment, Fleming 2017 should have been cited with respect to wake redirection only. The way it was written can easily cause confusion. Specific parts of that section will be rewritten to incorporate the mentioned source as well the mention of the Fleming source.

**Action:** The changes have been implemented as follows: "Induction control has shown great potential in different simulation and optimization environments (Marden et al., 2013; Ciri et al., 2017; Bossanyi et al., 2022). However, wind tunnel experiments and full-scale experiments have shown that the gain of induction control is negligible (Campagnolo et al., 2016; van der Hoek et al.,2019), contrary to what is found using wake redirection (Campagnolo et al., 2016; Fleming et al., 2017). The work described in Bossanyi et al. (2022) is currently being tested in a real-world wind farm to evaluate the effectiveness of their proposed induction control solution and could potentially yield a different conclusion to Campagnolo et al. 2016 and van der Hoek et al. 2019."

**Comment:** *Page 8: Figure 2: Recommend you identify some of the symbols in the figure in the legend for quick readability (Theta_Col -> Collective Pitch) St -> Strouhol number)*

*Figure 2: I associate the strouhol number with atmospheric properties, so wasn't sure why there would be a connection to mechanical frequency responses. Are these shown in Fig 2 just to show where these frequencies may, (coincidentally?), align?*

**Response:** The symbols will be clarified in the caption. The vertical dotted lines indicate which frequency aligns with certain Strouhal numbers given the wind turbine and inflow velocity. They are added for the reader to provide perspective which range of dynamics are of interest when using these blade pitching techniques.

**Action:** The following sentence has been added to caption of Fig. 2: "The vertical dotted lines indicate where blade pitching frequencies at 3 different Strouhal numbers would align with this data. For example, pitching at a frequency of St=0.50 for the NREL 5 MW turbine also means that the system is excited close to the eigenfrequency of the platform pitch motion. In the y-axes label $\theta_{col}$ refers to the collective pitch angle of the blades."